# On wake modeling, wind-farm gradients and AEP predictions at the Anholt wind farm

Alfredo Peña[1], Kurt Schaldemose Hansen[1], Søren Ott[1], and Maarten Paul van der Laan[1]

[1]DTU Wind Energy, Technical University of Denmark, Roskilde, Denmark

*Correspondence to:* Alfredo Peña (aldi@dtu.dk)

**Abstract.**

We investigate wake effects at the Anholt offshore wind farm in Denmark, which is a farm experiencing strong horizontal wind-speed gradients because of its size and proximity to land. Mesoscale model simulations are used to study the horizontal wind-speed gradients over the wind farm. From analysis of the mesoscale simulations and SCADA, we show that for westerly flow in particular, there is a clear horizontal wind-speed gradient over the wind farm. We also use the mesoscale simulations to derive the undisturbed inflow conditions that are coupled with three commonly-used wake models; two engineering approaches (the Park and G. C. Larsen models) and a linearized Reynolds-averaged Navier-Stokes approach (Fuga). The effect of the horizontal wind-speed gradient on annual energy production estimates is not found to be critical compared to estimates from both the average undisturbed wind climate of all turbines' positions and the undisturbed wind climate of a position in the middle of the wind farm. However, annual energy production estimates can largely differ when using wind climates at positions that are strongly influenced by the horizontal wind-speed gradient. When looking at westerly flow wake cases, where the impact of the horizontal wind-speed gradient on the power of the undisturbed turbines is largest, the wake models agree with the SCADA fairly well; when looking at a southerly flow case, where the wake losses are highest, the wake models tend to underestimate the wake loss. With the mesoscale-wake model setup, we are also able to estimate the capacity factor of the wind farm rather well when compared to that derived from the SCADA. Finally, we estimate the uncertainty of the wake models by bootstrapping the SCADA. The models tend to underestimate the wake losses (the median relative model error is 8.75%) and the engineering wake models are as uncertain as Fuga. These results are specific for this wind farm, the available dataset, and the derived inflow conditions.

## 1 Introduction

The Anholt wind farm is currently the fourth largest offshore wind farm in the world power-wise. The layout of the Anholt wind farm was optimized to minimize wake losses. The number of wind turbines (111), the wind-turbine type and the maximum allowed wind-farm area for turbine deployment (88 km$^2$) are examples of chosen constraints. The employed optimization tool has a tendency to place most wind turbines at the edges of the wind-farm area, while the remaining wind turbines are placed inside the wind farm with a relative large interspacing. For the particular case of Anholt, a number of wind turbines

were relocated from the optimized layout due to seabed that turned to be too soft (Nicolai Gayle Nygaard, 2017, personal communication).

So far the only reported studies on the wake effects of this wind farm are those of Nygaard (2014), Nygaard et al. (2014), and van der Laan et al. (2017). In the former, there is a comparison between the Park wake model (Katic et al., 1986) and SCADA for a row of turbines in the middle of the wind farm for a given wind-direction and wind-speed range. The wake model estimates the wake losses fairly well. The study also presents the results of the Park model for other large offshore wind farms, clearly showing that this wake model agrees with the SCADA for different inflow conditions rather well. These are interesting findings because engineering wake models do not generally include coupling with the vertical structure of the atmospheric boundary layer, thus, they should tend to underpredict wake losses in large offshore arrays (Stevens et al., 2016). However, the studies showing wake-model underprediction in large offshore wind farms (e.g., Barthelmie et al., 2009) analyze the wake observations using narrow wind direction sectors and do not account for wind direction variability. In the study by Nygaard et al. (2014), a comparison of two wake models, Park and the eddy viscosity model of WindFarmer (GL Garrad Hassan, 2013), is performed against SCADA, revealing that Park, with a wake-decay coefficient $k = 0.04$, gives better results than the model of WindFarmer with and without a large wind-farm correction. In the study by van der Laan et al. (2017), the effect of the coastline on the wind farm is investigated with a Reynolds-averaged Navier-Stokes (RANS) model, showing that such RANS setup is able to predict the horizontal wind-speed gradient over the wind farm when compared to the SCADA and mesoscale model simulations.

Engineering wake models are also often regarded as too simplistic for the estimation of wake losses, yet they are those that are most used when planning wind-farm layouts and for annual energy production (AEP) estimations. This is because they can be easily implemented and optimized in terms of computational performance. One cannot expect to characterize wakes in detail with such models but for the estimation of power and energy production means, they are sufficiently accurate when used properly (Nygaard, 2014; Nygaard et al., 2014). Peña et al. (2014) show that the Park model is able to predict the wake losses of the Horns Rev I wind farm in the North Sea for different atmospheric stability conditions when using a stability-dependent wake-decay coefficient. Peña et al. (2016) show that the Park model is in good agreement with the Sexbierum cases where two more sophisticated wake models are also tested: a linearized RANS solution (Fuga) and a nonlinear solution of the RANS equations that uses a modified $k$-$\varepsilon$ turbulence model. In the latter two studies, the high accuracy of the Park model is partly a result of accounting for the variability of the wind direction (Gaumond et al., 2014). Since Fuga is a computationally efficient wake model, whose results (in terms of wind-speed deficits) are nearly equal to those of a nonlinear solution of the RANS equations (Ott et al., 2011), we want to find out how different AEP and capacity factor estimates are when compared to those of Park and of another wake model that is a simple solution of the RANS equations, the G. C. Larsen model (Larsen, 2009).

Wake models of all types have been mainly evaluated against offshore wind farms that are well off the coast or where the effect of the land is assumed to be minimal (Barthelmie et al., 2009; Réthoré et al., 2013; Stevens et al., 2016). The layout of the Anholt wind farm offers the possibility of investigating the effect of land proximity ($\approx$20 km in the predominant wind direction) on the wind-farm production. We are aware that the Anholt wind farm experiences strong horizontal wind-speed gradients, which are translated into power gradients for turbines that are not experiencing wakes (Damgaard, 2015). Another

example of the effect of the land on an offshore wind farm, in this case in the Baltic Sea, is provided by Dörenkämper et al. (2015). The challenge is therefore to find out how such gradients interfere with the wake losses and how these affect the production and the AEP. This can be performed by simple 'coupling' of undisturbed[1] wind climates at some (or all) turbines' positions, in which the horizontal wind-speed gradient is embedded, with the wake models. To the authors knowledge, there have not been attempts to study the impact of the horizontal wind-speed gradient on wakes of wind farms using engineering wake models yet, although there is an attempt to include wind-direction gradients (Hasager et al., 2017). An obvious choice to derive the wind climate is the use of a mesoscale model such as the Weather Research and Forecasting (WRF) model (Skamarock et al., 2008), which is nowadays often used multi-purposely in the wind-energy community (Storm and Basu, 2010; Hahmann et al., 2015; Platis et al., 2018). In the present work, we also want to investigate the ability of WRF to model the horizontal wind-speed gradient over the wind farm.

In this study, we first present (Sect. 2) a general background regarding the Anholt wind farm, the WRF mesoscale runs that we use to estimate the wind-farm climate, the wind-farm SCADA, the wake models, and the ways in which we account for the horizontal wind-speed gradient and estimate the wake-models uncertainty. Section 3 presents the results regarding the influence of the wind-speed gradient on flow cases, on the AEP, those showing the evaluation of the wake models for two flow cases, and the analyses of the capacity factor, power loss and model uncertainty. Finally, discussion and conclusions are given in the last two sections.

## 2  Methods

### 2.1  Definitions

We define the efficiency of the wind farm at a given wind speed $U$ as

$$\eta_U = \frac{\sum_i P_i}{n_t P_U},$$  (1)

where $P_i$ the power of each individual turbine in the farm, $P_U$ the power of the turbine from the power curve at $U$, and $n_t$ the number of turbines in the wind farm.

We define the power loss of the wind farm as

$$PL = 1 - \frac{\langle \sum_i P_i \rangle}{n_t \langle P_{free} \rangle},$$  (2)

where $\langle \rangle$ means ensemble average and $P_{free}$ is the power of the free-stream turbines (these are defined in Sect. 2.2.2).

We define the relative wake model error as

$$\epsilon = \frac{PL_{obs} - PL_{mod}}{PL_{obs}},$$  (3)

where the subscript $_{obs}$ and $_{mod}$ refer to observations and model, respectively.

---

[1]undisturbed is in this study referred to as a wake-free condition

## 2.2 Anholt wind farm

The Anholt wind farm is located in the Kattegat strait between Djursland and the island of Anholt in Denmark (Fig. 1-left). It consists of 111 Siemens 3.6 MW-120 turbines with hub height of 81.6 m and a rotor diameter of 120 m (Fig. 1-right). The smallest distance between the turbines is 4.9 rotor diameters. The water has depths of 15–19 m, the wind farm area is 88 km$^2$ and full operation started in summer 2013.

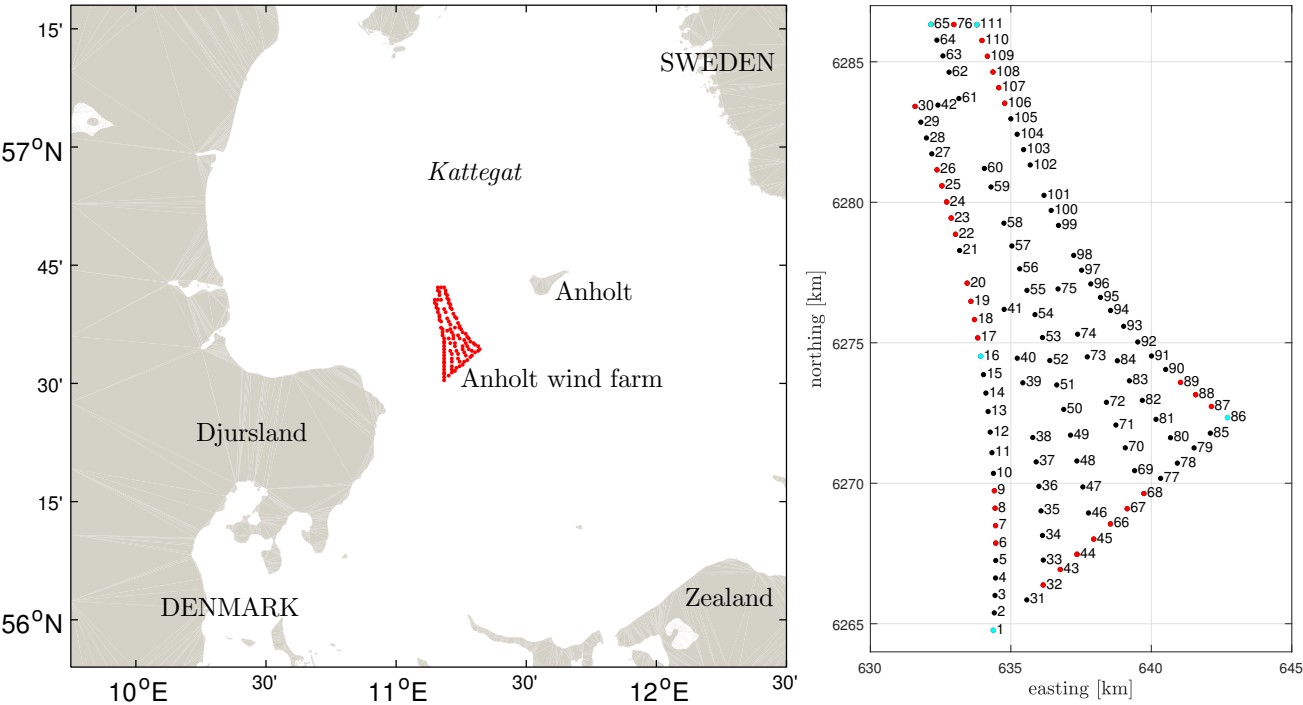

**Figure 1.** (Left) the Anholt wind farm (red markers) in the Kattegat. (Right) The layout and numbering of the turbines of the Anholt wind farm in UTM32 WGS84. Turbines used to derived the inflow conditions are shown in red and cyan markers

### 2.2.1 SCADA

We have access to 10-min means of SCADA for the period January 1, 2013 to June 30, 2015. Data include nacelle wind speed, yaw position, pitch angle, rotor speed, power reference, air temperature, rotor inflow speed, and active power. We also produce a filtered SCADA dataset by identifying periods where each turbine was grid connected and produced power during the entire 10-min period. The dataset excludes periods where any turbine was either parked or idling, those with starting and stopping events, where power was curtailed, or boosted. We find turbines nr. 1, 36, 65, and 68 to be boosted with power values 5% above the rated value. The result is a time series of 7440 10-min values starting in July 2013 until December 2014.

### 2.2.2 Inflow conditions

Due to the lack of undisturbed mast measurements in the SCADA, we derive the inflow conditions from the filtered SCADA dataset. We estimate an 'equivalent' wind speed based on either the 10-min SCADA's power or pitch angle values in combination with the manufacturer's power curve or the average pitch curve extracted from the SCADA. The inflow reference wind speed is computed as the average equivalent wind speed for groups of four undisturbed turbines as shown in Table 1. A group of four turbines is used to robustly estimate the inflow wind speed and 10 different sectors are needed to avoid the influence of Djursland and the island of Anholt. The inflow reference wind direction is computed as an average yaw position for pairs of undisturbed wind turbines listed in Table 1. The yaw position calibration is performed as in Rodrigo and Moriarty (2015). The turbines that we use to derive the inflow conditions are shown in Fig. 1-right.

**Table 1.** Free-stream turbines used to determine the inflow wind speed (first two columns) and the inflow wind direction (second two columns) as function of an average yaw position

| yaw [deg] | turbine nr. | yaw [deg] | turbine nr. |
|-----------|-------------|-----------|-------------|
| 0–35 | 65 76 110 111 | 0–30 | 65 111 |
| 35–55 | 106 107 108 109 | 30–90 | 111 86 |
| 55–90 | 86 87 88 89 | | |
| 90–180 | 45 66 67 68 | 90–210 | 1 86 |
| 180–215 | 32 43 44 45 | | |
| 215–230 | 6 7 8 9 | 210–330 | 1 16 |
| 230–270 | 22 23 24 25 | | |
| 270–280 | 17 18 19 20 | | |
| 280–310 | 23 24 25 26 | | |
| 310–360 | 30 65 76 111 | 330–360 | 65 111 |

## 2.3 Wind-farm climate

We perform simulations of the wind climate over a region covering the Anholt wind farm using the WRF version 3.5.1 model. Simulations are carried out on an outer grid with horizontal spacing of 18 km $\times$ 18 km (121 $\times$ 87 grid points), a first nested domain of 6 km $\times$ 6 km (280 $\times$ 178 grid points), and a second nest with center in the middle of Jutland, Denmark of 2 km $\times$ 2 km (427 $\times$ 304 grid points). The simulations use 41 vertical levels from the ground to about 20 km. The lowest 12 levels are within the 1000 m of the surface with the first level at $\approx$14 m. Initial, boundary conditions, and fields for grid nudging come from the European Centre for Medium Range Forecast ERA-Interim Reanalysis (Dee et al., 2011) at $0.7° \times 0.7°$ resolution. Other choices in the model setup are standard and commonly used in the modelling community. Further details regarding the simulations are provided in Peña and Hahmann (2017). Figure 2 shows the Anholt wind climate at hub height at a WRF grid point in the middle of the wind farm based on the WRF hourly outputs for 2014 (the model is run for 1982–2015). The model

output is logarithmically interpolated to hub height. Most winds come from the west, south-southwest, and southeast directions and winds between 5 and 15 m s$^{-1}$ are the most frequent (the all-sector mean wind speed is 9.23 m s$^{-1}$ at hub height).

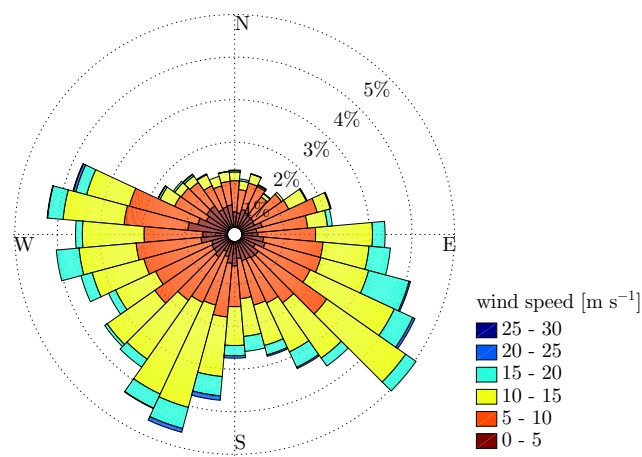

**Figure 2.** The wind climate at hub height in the middle of the Anholt wind farm for the year 2014 based on WRF simulations

## 2.4 Wake models

We use three different wake models: the Park wake model with the commonly-used offshore value of $k = 0.04$, the G. C.
Larsen model (Larsen, 2009), and Fuga (Ott et al., 2011). The first two are engineering wake models and Fuga is a linearized flow solver of the steady-state RANS equations using an actuator-disk approach. For the two engineering wake models, the local wake deficits $\delta_i$ are superposed to compute the speed deficit at the $n$th turbine. This is performed in two different ways: linearly $\sum_{i=1}^{n} \delta_i$ and as a quadratic sum $\left(\sum_{i=1}^{n} \delta_i^2\right)^{1/2}$.

Due to the high computational efficiency of these wake models, we can easily perform wake analyses over given wind-speed and wind-direction ranges and AEP-like calculations using the values in the time series (no need for distributions). For the latter calculations, we create look-up-tables (LUTs) for each wake model, which contain the total wind-farm power output for specific undisturbed wind directions and wind speeds. Figure 3 shows a comparison of the efficiency of the wind farm (Eqn. 1) predicted by the wake models. All wake models show the highest wake losses at the directions where most wind turbines are aligned, i.e. at $\approx$160 and 340 deg, and 45 and 235 deg. At 5 m s$^{-1}$, the Park-linear model generally shows the highest wake losses followed by Larsen-linear and Fuga (within the direction where turbines are most aligned). At 5 and 10 m s$^{-1}$, $\eta \approx 0.9$ for all wake models excluding the most aligned directions, being Larsen-quadratic and Park-linear the models showing the highest and lowest efficiencies, respectively.

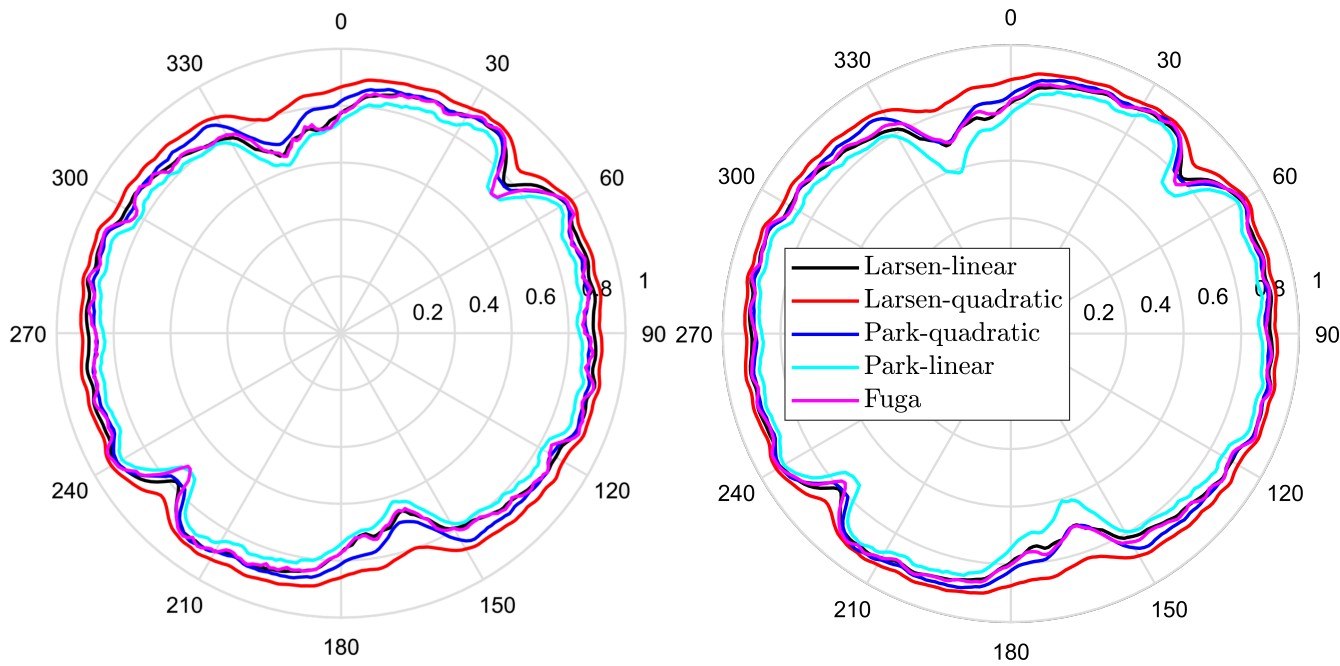

**Figure 3.** The efficiency of the Anholt wind farm predicted by the wake models at 5 m s$^{-1}$ (left frame) and 10 m s$^{-1}$ (right frame)

## 2.5 Accounting for the wind-farm gradient

One way to account for the effect of the horizontal wind-speed gradient within a wind farm, which is not the result of wake effects themselves, on the wind-farm power output is by estimating the wake losses using the undisturbed wind speed and direction at each individual turbine position for each time realization as inflow condition instead of using a single undisturbed wind speed and direction as it is commonly performed. At each turbine position, we will therefore have both a time series of velocity deficits (and thus power values) because of the change with time of inflow conditions and a series, with a number of members equal to the number of turbines in the farm, of velocity deficits for each inflow condition experienced by each turbine for each time realization. Then, the wind-farm power time series, as an example, can be estimated by averaging the power resulting from all inflow conditions for the same time realization (for the Anholt case this means 111 conditions) and then averaging the results of all turbines. This is hereafter known as a gradient-based analysis. The wind/inflow at each turbine must be undisturbed and so mesoscale model simulations over the wind-farm area (without the wind farm) are an obvious option to estimate the wind climate at each turbine position.

Due to the very high efficiency of the Park model (in a Matlab script it takes miliseconds to perform one simulation of Anholt for a single inflow wind speed and direction), when using the WRF hourly time series, we can perform 111 simulations (i.e., 111 different inflow conditions that are interpolated from the WRF grid into the turbine positions) in a couple of seconds. Thus, we can perform a gradient-based AEP analysis with hourly WRF winds in just few hours. It is important to note that we

can perform traditional (i.e., with a single inflow condition per time realization) AEP calculations with all wake models much faster using pre-computed LUTs.

## 2.6 Uncertainty estimation

We quantify the uncertainty of the wake models using a nonparametric circular-block bootstrap similar to the approach of Nygaard (2015). The idea is to 'wrap' the power-output time series (from both measurements and simulations) of the wind farm around a circle. Blocks of the time series with a given size, which is here selected according to Politis and White (2004) based on the wind-speed time series, are then randomly sampled. The number of sampled blocks is given by the total size of the time series and the block size. The number of bootstrap replications should be large enough to ensure a close to zero Monte Carlo error. By bootstrapping the power-output time series, we can estimate the bootstrapped $PL$ (Eqn. 2) and so estimate a distribution of $\epsilon$ (Eqn. 3). Details and code implementations of a number of bootstrapping techniques can be found in Sheppard (2014).

## 3 Results

The analysis of the influence of the horizontal wind-speed gradient in Sect. 3.1 is performed with the WRF model outputs for 2014 and the filtered SCADA dataset. For AEP estimations (Sect. 3.1.1), we only use WRF model outputs. The westerly flow case in Sect. 3.1.2 uses the filtered SCADA dataset, as well as the south flow case in Sect. 3.1.3, and the WRF model outputs. For the capacity factor calculations in Sect. 3.2, we use all the SCADA available for 2014 and the WRF model outputs for the same year. The analyses of the power loss and model uncertainty in Sects. 3.3 and 3.4 are performed on the filtered SCADA.

## 3.1 Influence of the wind-farm gradient

Figure 4 shows the mean horizontal wind-speed gradient at hub height in and surrounding the Anholt wind farm based on simulations from the WRF model for the year 2014. The left frame shows the average for all wind speeds and directions and the right frame the average for all wind speeds and directions within $270 \pm 30$ deg, which have been filtered using the simulated wind direction at hub height at the position of turbine 15. The influence of Djursland (see Fig. 1-left) on the wind at the farm is clear even for the omnidirectional case. The impact of Djursland is much stronger when looking at westerly winds so we could expect an impact on the results of wake models when the flow is particulary from these directions. The horizontal wind-speed gradient is mainly due to the roughness effect of the land surrounding the wind farm (van der Laan et al., 2017). Although it is not shown, the island of Anholt east of the farm also has an impact on the wind speed at the wind farm for northeasterly flow but this is not as strong as that of Djursland for westerly flow. For westerly winds ($270 \pm 30°$), the WRF-simulated average hub-height wind-speed difference between turbines nr. 1 and 30 is 0.62 m s$^{-1}$, whereas for easterly winds ($90 \pm 30°$) it is 0.12 m s$^{-1}$ between turbines nr. 86 and 111.

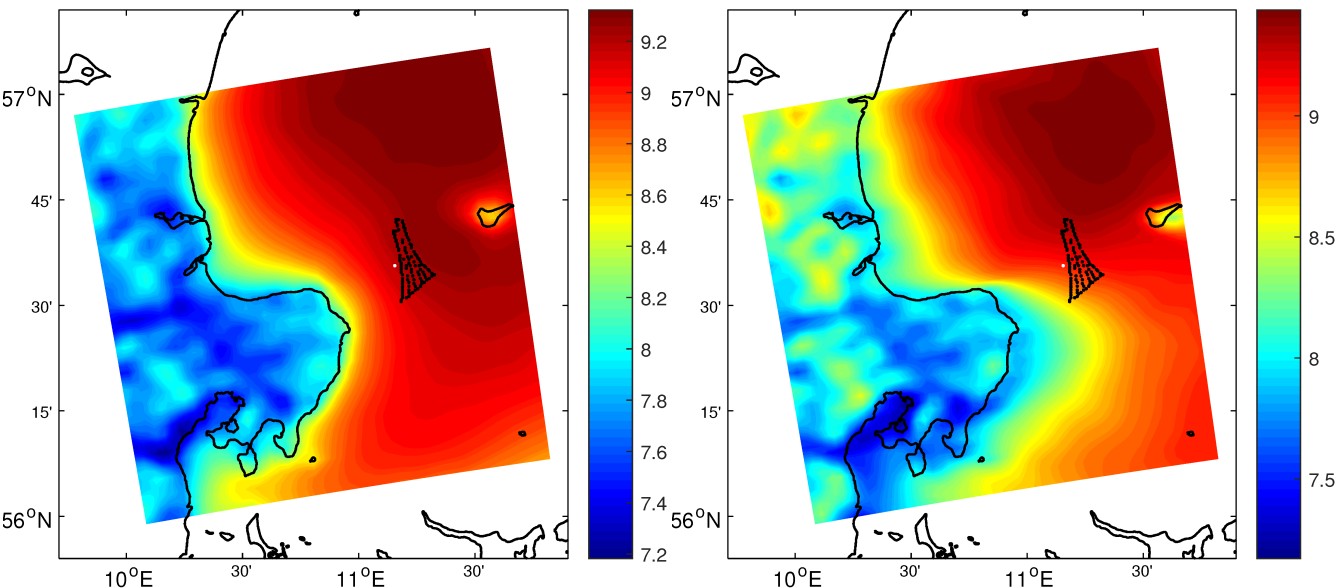

**Figure 4.** WRF-simulated mean wind speed at hub height on the Kattegat area where the Anholt wind farm is deployed for the year 2014. All data are shown in the left frame and data within the directions $270 \pm 30°$ at the position of turbine nr. 15 in the right frame. Colorbars are in m s$^{-1}$

In Fig. 5-left we extract the values from Fig. 4 at each turbine position by linearly interpolating the WRF winds to the turbine positions. For the omnidirectional case, the horizontal wind-speed gradient is lower than for westerly winds, as expected, and for both cases the strongest gradient is observed for the first row of turbines (1–30), which are those closer to Djursland.

Figure 5-right shows SCADA-derived and WRF-simulated average wind speeds at hub height for turbines nr. 1–30 for a number of westerly flow cases. We select filtered SCADA based on the inflow conditions described in Sect. 2.2.2 within the wind-speed range 5–10 m s$^{-1}$ and use the manufacturer's power curve to derive each turbine's wind speed from the power output. For the comparison, we extract the WRF-simulated winds by averaging the horizontal wind-speed components on the corresponding free-stream turbines for each direction range as given in Table 1. We also select WRF-simulated winds within the same wind-speed range 5–10 m s$^{-1}$. It is observed that the horizontal wind-speed gradient for westerly winds depends on the particular direction. The strongest simulated and observed gradients are found at $265 \pm 5$ deg, being the winds at turbines nr. 1–15 lower than those at turbines nr. 15–30. Generally, the simulated gradient agrees with the observations fairly well, except for the range $295 \pm 5$ deg, where the SCADA show the highest winds at the southern turbines. This can be an effect of the topography on the turbines, which is not captured by WRF. It could also be a wind-farm wall effect (Mitraszewski et al., 2012). A similar effect (not shown) is observed when analyzing the SCADA-derived wind speeds of the turbines at the south of each row for a direction 80–90 deg: the wind speed at turbine nr. 1 is about 6% higher than that at turbine nr. 86.

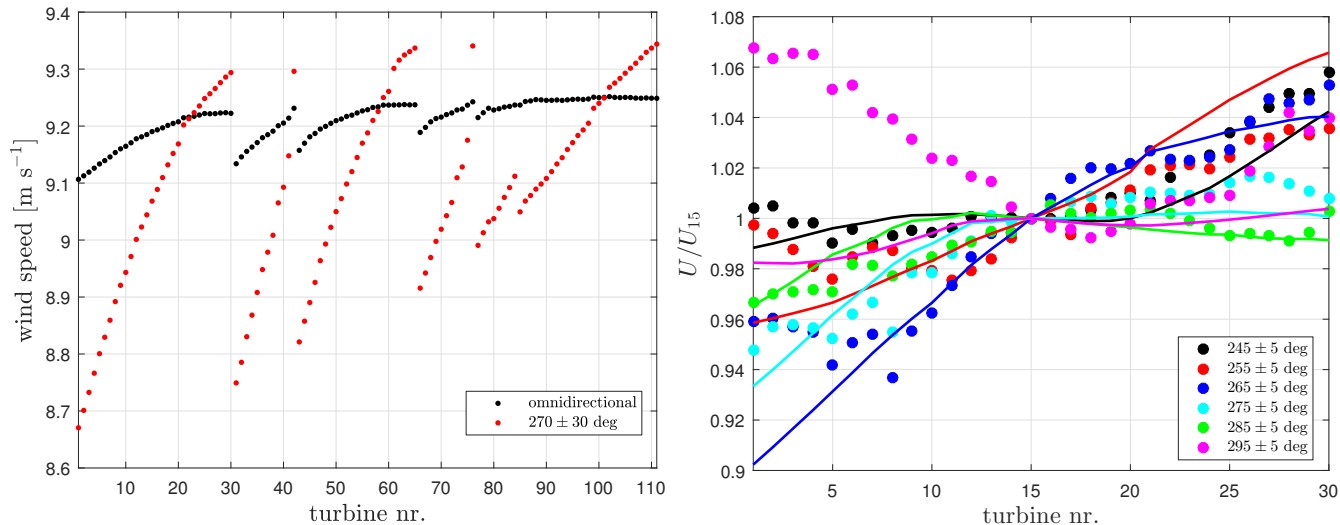

**Figure 5.** (Left) WRF simulated average wind speed at hub height at the turbine positions of the Anholt wind farm. (Right) Average wind speed at hub height (normalized by that of turbine 15) at the most westerly row of the wind farm for a number of westerly flow cases: WRF winds in solid lines and SCADA-derived winds in markers

### 3.1.1 Annual energy production

The difference in AEP when accounting for the wind-farm gradient information and when assuming a horizontally homogenous wind field[2] is lower than 1% when using the 2014 hourly WRF wind fields combined with the wake models ('average wind field' column in Table 2). This is because, in general for this wind climate, there are positive/negative errors in the production estimations that are balanced during the year. The highest difference is observed for the WRF-Fuga setup, in which the estimation using the 'average wind' does not balance for the low energy yield of the turbines in the south of the farm and the high energy yield of those in the north as it does for the other WRF-wake model setups.

**Table 2.** Difference (in percentage) between different type of AEP calculations and that using the horizontal wind-speed gradient information from the WRF simulations

| model setup | average wind field | turbine nr. 1 | turbine nr. 54 | turbine nr. 65 |
|---|---|---|---|---|
| WRF-Park-quadratic | 0.05 | -1.29 | 0.08 | 0.26 |
| WRF-Park-linear | 0.05 | -1.33 | 0.07 | 0.26 |
| WRF-Larsen-linear | 0.05 | -1.28 | 0.08 | 0.27 |
| WRF-Larsen-quadratic | 0.06 | -1.24 | 0.08 | 0.27 |
| WRF-Fuga | 0.76 | -0.59 | 0.77 | 0.98 |

[2]estimated each hour by taking the average of the horizontal wind-speed gradient over each turbine of the farm

The difference in the AEP estimation by accounting for the wind-speed gradient and that by using the wind climate of turbine nr. 1, which is the position with the lowest average wind speed, is larger than 1% for the engineering wake models. Such difference is rather large considering that the AEP of the wind farm is ≈1889.3 GW h when averaging all models' AEP estimations using the wind-gradient information. The same exercise using the information of turbine nr. 54 (in the middle of the farm) results in differences very close to those using the average wind field. Using the information of turbine nr. 65 (at the top of the farm), the difference is also large but positive as expected. For the Anholt wind farm and its wind climate, in particular, these results show that although accounting for the wind-farm gradient is important, it does not change largely the AEP estimations compared to those based on a one-point wind climate, unless the latter is not close to the average wind climate within the wind-farm area. For comparison purposes (e.g. with the results in Fig. 5-left) the yearly average wind speed of the 'homogenous' wind is 9.21 m s$^{-1}$.

### 3.1.2 Westerly flow cases

Given the impact of the horizontal wind-speed gradient on the AEP estimations (Sect. 3.1.1), it is relevant to study the wake losses under westerly flow conditions. Figure 6-top shows, for 2014, the average WRF-Park-quadratic power of each turbine in the wind farm when filtering for westerly wind directions (using the WRF simulated wind climate at turbine nr. 15), both accounting for the wind-speed gradient, as described in Sect. 2.5, and assuming a homogenous wind field (the average of the wind climates at each turbine). For a broad wind-direction range, both results are nearly identical and only small differences at specific turbines (up to 27.2 kW) are found when the wind-direction range is reduced; in this latter case we use the range that shows the largest gradients in Fig. 5-right. It is important to note that, although it is not seen, the normalized average power of turbines 1–30 for the two 'gradient' cases in Figure 6-top is slightly lower than one as expected.

Since the horizontal wind-speed gradient does not seem to strongly impact the wake behavior for broad wind-directions ranges, we compare the SCADA that have been wind-speed and direction filtered with the wake models in Fig. 6-bottom. The inflow conditions are derived from the SCADA (see Table 1) and are used to run the wake models. 735 10-min cases are left after filtering for wind speed and direction (5–10 m s$^{-1}$ and $270 \pm 30$ deg). In this case the power values are not normalized with the power of a unique turbine, as we do for the plot in the top frame. Instead, we use the undisturbed turbine that is closest to that where we are extracting the power from. This aids to levelize the SCADA at turbines nr. 1–30 mainly. The wake models generally agree with the SCADA, particularly Fuga, being this and the engineering wake models' variants using the linear sum of wake deficits those showing the highest wake losses generally. For turbines nr. 31–60, where the wind farm experiences single and double wakes mostly, the SCADA are between the models' results. For turbines nr. 66–111, where multiple wakes occur, Larsen-quadratic highly underestimates the wake and the linear 'variants' and Fuga seem to generally agree better with the SCADA. However, the comparison is not completely fair with the wake models because the reference power is not always higher or equal to that of the individual turbines when these are supposed to be in the wake of a turbine. E.g. in the case of turbine nr. 31, we use turbine nr. 3 as reference and in ≈19% of the cases with the inflow conditions analyzed in Fig. 6-bottom, $P_3 < P_{31}$.

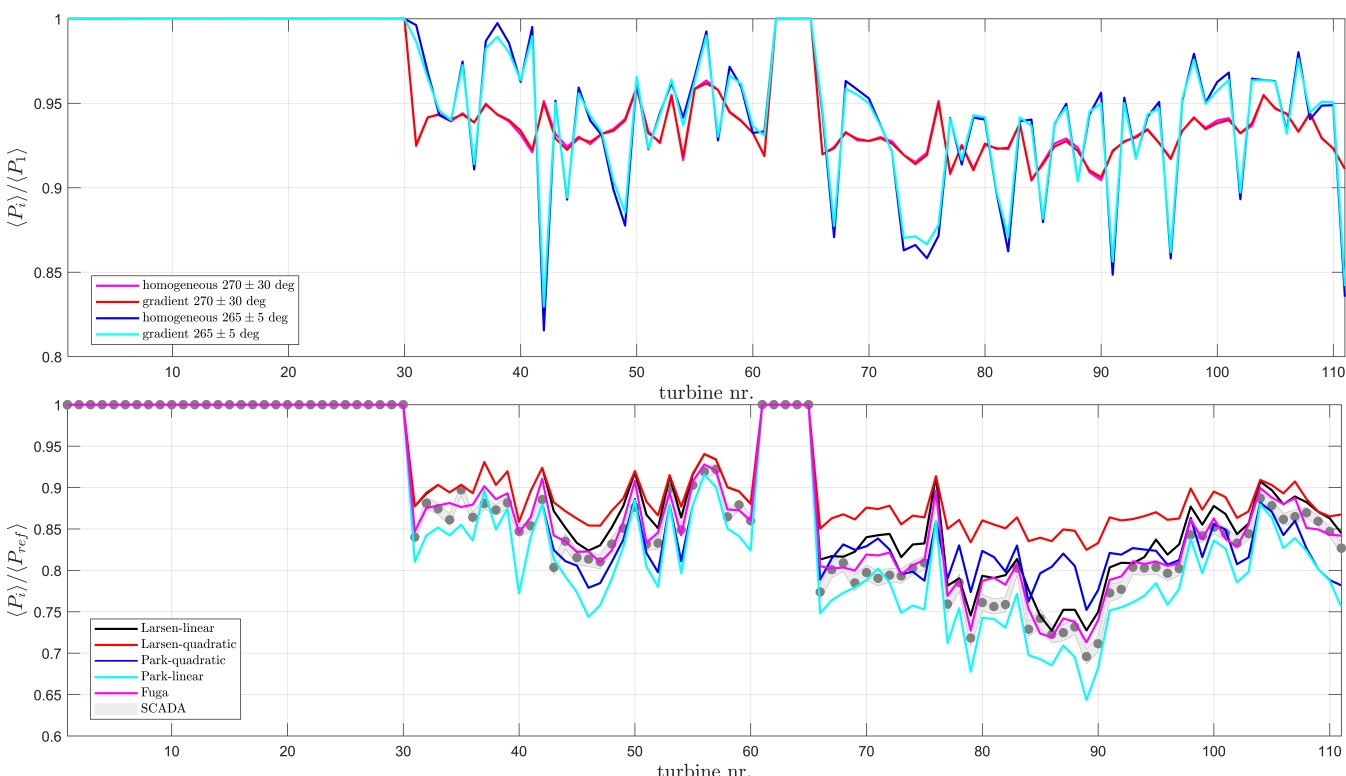

**Figure 6.** Normalized average power of each turbine in the wind farm for westerly flow conditions. (Top) from simulations using the 2014 WRF time-series and Park-quadratic with (gradient) and without (homogeneous) the horizontal wind-speed gradient information. (Bottom) from SCADA and simulations from the wake models within the range $270 \pm 30$ deg and hub-height inflow wind speed of 5–10 m s$^{-1}$. For the SCADA, the shaded region indicates $\pm$the standard error of the mean

### 3.1.3 Southerly flow case

Figure 7 illustrates the wake loss for the north-south row in the middle of the wind farm (turbines nr. 45–65) filtering for inflow conditions ($9 \pm 0.5$ m s$^{-1}$ and $168.7 \pm 15$ deg, which is the direction where turbines nr. 45 and 46 are aligned) that are derived from the SCADA of turbines nr. 45 and 66–68 (Table 1). 26 10-min cases are left after filtering for wind speed and direction.

5   As expected from the results in Fig. 6-bottom, for this multiple wake case, the models using the 'linear' variant agree better with the SCADA than those using the 'quadratic' variant when going deeper in the row. The Park-quadratic model predicts the wake loss of the three first turbines rather well but underpredicts it when moving deeper in the row. The results from Fuga are between the engineering model's variants.

   Because the differences between SCADA and models in Fig. 7 are relatively large and the amount of 10-min periods for

10   the southerly flow case are 26 only, we also perform actuator-disk RANS simulations in EllipSys3D (Sørensen, 2003) using a modified $k$-$\varepsilon$ turbulence model (van der Laan et al., 2015). The results of the RANS model are very close to those of

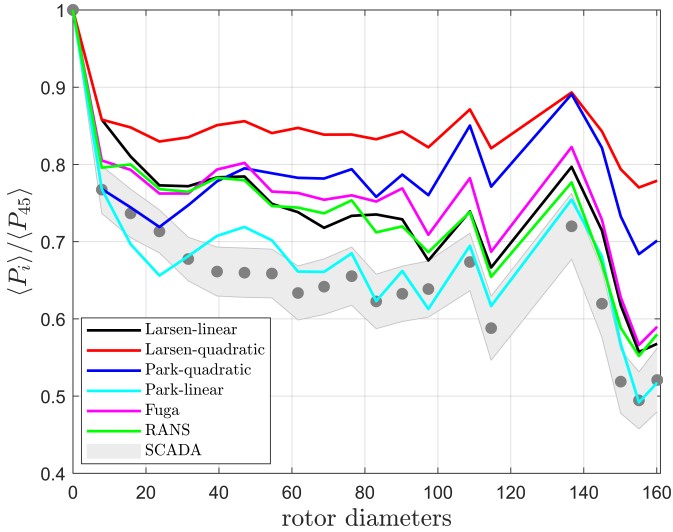

**Figure 7.** Normalized average power of the north-south row of turbines in the middle of the wind farm for southerly flow conditions from SCADA and simulations from the wake models within the range $168.7 \pm 15$ deg and hub-height inflow wind speed of $9 \pm 0.5$ m s$^{-1}$. For the SCADA, the shaded region indicates $\pm$the standard error of the mean

Fuga and Larsen-linear also underestimating the wake loss. We can only speculate that for this particular case, the high wake loss from the SCADA is due to atmospheric conditions, in particular from periods under a rather stable atmosphere, that we are not accounting for in the simulations. However, we do not have useful observations to directly derive stability. We have atmospheric stability measures from the WRF simulations but 'instantaneous' WRF stability measures are highly uncertain (Peña and Hahmann, 2012). Nygaard (2014) shows the same case using another SCADA period and the wake losses are $\approx$10% lower than those we observe.

## 3.2 Capacity factor

Being able to estimate the AEP (Sect. 3.1.1) is important but it is more interesting to find out whether we are able to predict it, in our particular case, with the combined mesoscale-wake setup. For the exercise, the capacity factor is a better choice than the AEP, since we can compare Anholt with other offshore wind farms.

We use all the SCADA data that are available for 2014. Theoretically, there should be 52560 10-min samples for this year. However, the amount of samples per turbine available in the SCADA varies and is never the theoretical one; the turbine with the highest amount of samples is nr. 7 (51648) and that with the lowest is nr. 77 (49512). The average availability, taking into account all turbines, of observed samples is 98.10%. Table 3 shows the observed and estimated capacity factors, which are predicted by the WRF-wake model setup and that account for both the wind-farm gradient and the observed average availability of samples.

**Table 3.** Observed and estimated (from the WRF-wake model setup) capacity factors of the Anholt wind farm for 2014. The estimated values account for the observed average availability of samples. The last column shows the power loss based on the SCADA and the power loss estimations from wake models without WRF coupling

| source | capacity factor [%] | power loss [%] |
|---|---|---|
| SCADA | 51.75 | 4.08 |
| WRF-Park-quadratic | 53.19 | 3.64 |
| WRF-Park-linear | 51.89 | 5.05 |
| WRF-Larsen-linear | 52.87 | 3.87 |
| WRF-Larsen-quadratic | 54.13 | 2.60 |
| WRF-Fuga | 52.51 | 3.70 |

It is clear that we can estimate fairly well the observed capacity factor using the WRF-wake model setup. However, it is important to note that wind turbines are not always working and underperform when compared to the manufacturer's power curve. The predicted AEP/capacity factor of a combined mesoscale-wake model is typically higher than the observed value; however, we want to know the capacity factor of a wind farm regardless of the operating conditions.

## 3.3 Power loss

Table 3 also shows the wind farm $PL$ based on the SCADA's 7440 10-min values and using Eqn. (2) with the inflow conditions as defined in Table 1. The results for the wake models are computed interpolating the models' LUTs with the same inflow conditions derived from the SCADA. All models, except for Park-linear, predict lower $PL$s than the SCADA; Park-quadratic, Larsen-linear and Fuga slightly underestimating the wake loss.

One way to show that the estimations of power of the free-stream turbines are sound is to compare the manufacturer power curve with the SCADA-derived power (averaging the power of the turbines in Table 1) and SCADA-derived inflow wind speed. This is illustrated in Fig. 8-left, where we show the power curve of the turbine and the SCADA-derived values (no interpolation is made). Figure 8-right shows a similar comparison but in this case we derive the gross wind-farm power (i.e. 111 times the power of the free-stream turbines) and that derived from the power curve at the estimated free wind speed. Both figures show that our definition of the free-stream turbines is sound (no evident wake effects are observed) and that the turbines do follow the manufacturer's power curve.

However, this does not give us an idea about the validity of the SCADA-derived inflow conditions for the turbines that are far from those we use to derive the inflow conditions. By filtering the SCADA-derived inflow conditions for westerly flow ($270\pm30$ deg), so that no wakes are observed for turbines nr. 1–30, we can derive power curves for the turbines at the beginning and end of that row (i.e. nr. 1 and 30) and compare them to, e.g. the manufacturer's power curve. As expected, the power curves for turbines nr. 1 and 30 are below and above the manufacturer's one, being the difference as high as 500 kW for turbine nr. 1, which is the turbine with the lowest average wind speed according to the WRF simulations (Fig. 5-left). Within the wind-speed range where we observe such differences in power, the difference in wind speed is about 1 m s$^{-1}$.

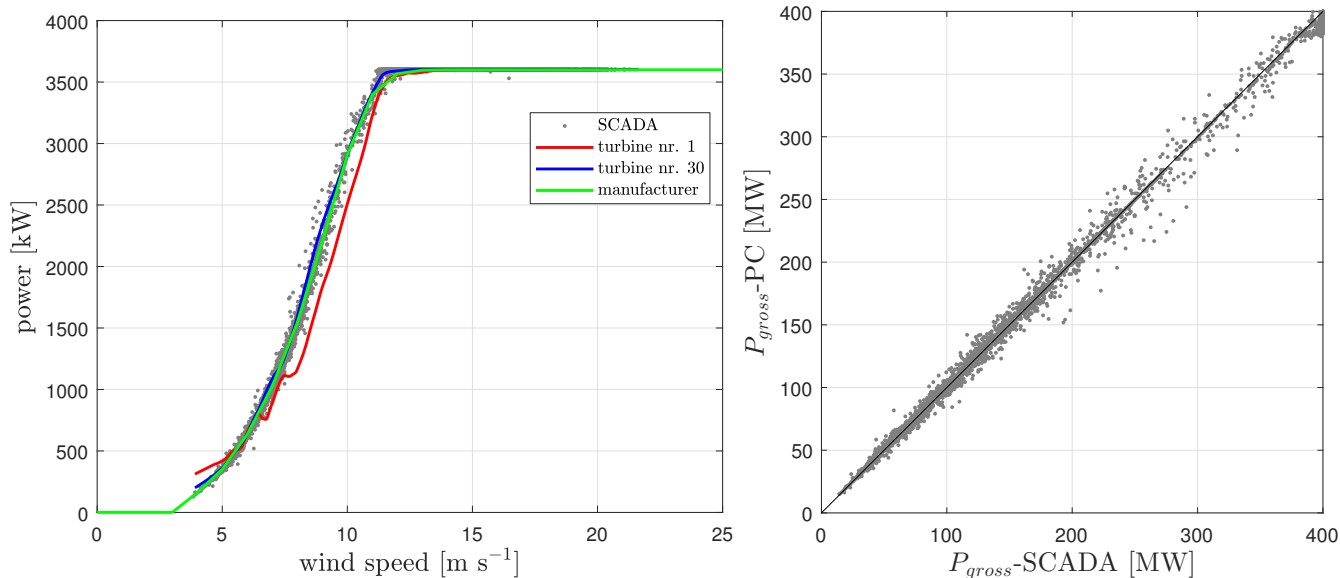

**Figure 8.** (Left) power curve of the turbines at the Anholt wind farm derived from the SCADA of free-stream turbines compared to the manufacturer power curve. (Right) gross wind-farm power derived from the SCADA for the free-stream turbines compared to that derived from the power curve (PC)

### 3.4  Model uncertainty

Also based on the SCADA's 7440 10-min values, we find an optimal block length for the circular bootstrap of 242 samples. In average, such sample length corresponds to about 10 days, which is long enough to capture the correlation between samples. We use 10000 bootstrap replications and find that, e.g. $\epsilon$ for the Park-quadratic model stabilizes after 2000 replications. Figure

9 shows the distribution of $\epsilon$ for all models where positive $\epsilon$ values denote a model that overestimates the power (underestimates the wake loss), whereas negative $\epsilon$ values a model that underestimates the power (overestimates the wake loss).

For the particular case of the Anholt wind farm and for the filtered SCADA used in the analysis, Larsen-linear has the distribution with lowest bias and the second largest $\sigma$ value (after Park-linear), whereas Larsen-quadratic has the highest bias and lowest $\sigma$ values. The results for Park-quadratic and Fuga are similar, both bias and $\sigma$. Park-linear, as expected due to the

previous results, is the only model systematically overestimating the wake loss. If we could extrapolate these results to an AEP analysis, we would expect non-conservative AEP estimations (except for Park-linear), being Park-quadratic, Fuga and Larsen-linear slightly optimistic and Larsen-quadratic too optimistic.

## 4  Discussion

It is important to note that some of our results depend on the methods we use to derive the undisturbed inflow conditions of

the wind farm. We show that for power analyses of individual turbines, whose inflow conditions are greatly affected by the

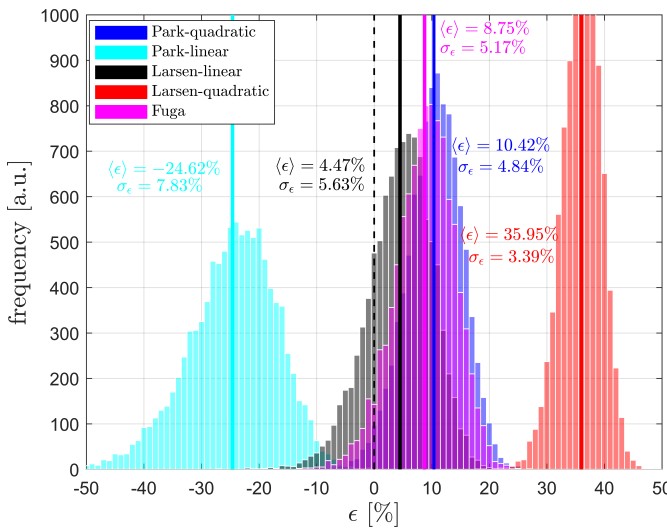

**Figure 9.** Distribution of the relative model error $\epsilon$ in estimating power losses (Eqn. 3) of three wake models using 7440 10-min bootstrapped samples from the Anholt wind-farm SCADA. The mean of each distribution is shown with a thicker vertical line. The mean and standard deviation of the distributions of $\epsilon$, $\langle\epsilon\rangle$ and $\sigma_\epsilon$, are also given

horizontal wind-speed gradient (like turbines nr. 1 or 30), this is an important matter (see Fig. 8-left). For this particular wind farm and wind climate, the differences between the undisturbed inflow conditions derived from turbines in the middle of the long rows and the inflow conditions derived from turbines to either side of the rows compensate for the overall wind-farm long-term analyses (e.g. AEP and capacity factor). One way to further analyze the impact of different inflow conditions is to
5 derive them for each individual undisturbed turbine. We can then potentially perform analyses (flow cases, power loss, and capacity factor) in a similar fashion as that we use for accounting for the horizontal wind-speed gradient[3] and validate our findings.

We also estimate the power loss and the uncertainty of the wake models based on a rather discontinuous and short filtered SCADA dataset. Therefore, our results might be biased and caution must be taken when generalizing our findings. A clear
example is that related to the model uncertainty where we find that most wake models underestimate the wake losses. With a longer dataset, the biases can change (and models might start to produce conservative results) but the relative position of the models will most probably be maintained, Park-linear and Larsen-quadratic being the most conservative and most optimistic models, respectively. If the same models are evaluated with SCADA from other wind farms, the biases will most probably change.
We show that our WRF-wake model setup is able to predict rather accurately the capacity factor of the Anholt wind farm. Anholt is the offshore wind farm with the highest all-life capacity factor in Denmark (48.7%) and the highest in the world

---

[3]although we cannot derive the undisturbed horizontal wind-speed gradient from wake-affected turbines without a wake model

for a wind farm older than 2 y, outperforming Horns Rev II that has in principle more favourable wind conditions. One of the reasons for this is the Anholt wind-farm layout, which highly minimizes the wake losses.

The results for the two flow cases illustrate what we already expected; Park-linear showing the highest and Larsen-quadratic the lowest wake deficits. This is mainly because of the values we choose for the wake decay coefficient. It is important to note that we can get similar wake deficits with both the Park-linear and Park-quadratic models when tuning the wake decays. Physically, it makes more sense to linearly sum the wake deficits but the quadratic approach is normally used due to a historical general good match of model predictions with observed power deficits, for the values normally suggested for the wake decay (0.04–0.05 for offshore conditions). The RANS model shows similar values to Fuga, as expected due to the similarity of the models' physics, both showing a better comparison to the SCADA for the two flow cases than the 'traditional' Park-quadratic model, also as expected.

## 5   Conclusions

For the Anholt wind farm, we show from both the SCADA and WRF model simulations that for a number of wind directions, there is a clear influence of the land on the free-stream wind speed at the positions of the turbines closer to the coast. However, for AEP calculations where we run three different wake models using mesoscale model outputs as inflow conditions, accounting for the horizontal wind-speed gradient (also derived from the mesoscale model results) does not have a large impact on the results when compared to AEP calculations based on first, a wind climate that is the average of all wind climates at the turbines' positions, and second, a wind climate correspondent to a position in the middle of the wind farm. It does, however, differ from the calculation using a wind climate that is strongly influenced by the horizontal wind-speed gradient particularly for the engineering wake models.

We look at two flow wake cases with two different engineering wake models and some of its variants and a linearized RANS model. The first case corresponds to westerly winds, where the influence of the horizontal wind-speed gradient is largest. Here the wake models, and Fuga in particular, agree with the SCADA fairly well. The second case corresponds to southerly winds, where the wake losses are highest. Here, the wake models tend to underestimate the wake deficit when compared to the SCADA. This is also translated into a wake-model tendency to underestimate the observed power loss; in average 0.31% less than that derived from the SCADA.

Using our mesoscale-wake model setup, we find that the estimated capacity factors are 0.27–4.60% biased when compared to that computed from the SCADA. Finally, using inflow conditions derived from the SCADA and by circularly block-bootstrapping these, we estimate the relative error of the wake models. We find that these models tend to underestimate the wake losses, except for one wake model variant. The engineering wake models are found to be as good as the linearized RANS fuga model. However, these are results that are wind-farm and SCADA specific, and that depend on the definition of inflow conditions; therefore similar analyses need to be reproduced at different wind farms, using more SCADA and different methods to derive the inflow conditions.

*Data availability.* The Anholt SCADA can be made available by Ørsted upon request to Miriam Marchante Jiménez (mirji@orsted.dk). The WRF data can be made available by DTU Wind Energy upon request to Andrea N. Hahmann (ahah@dtu.dk).

*Competing interests.* The authors declare that they have no conflict of interest.

*Acknowledgements.* We would like to thank Ørsted and partners for providing the SCADA. Also, we thank Charlotte B. Hasager to promote
and lead the Anholt wind-farm internal project at DTU Wind Energy and Patrick Volker to make the mesoscale model simulation outputs
easily accessible. Finally, we would like to thank the three anonymous reviewers and Nicolai Gayle Nygaard for their comments on the
manuscript.

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
