# Peer review of "On wake modeling, wind-farm gradients and AEP predictions at the Anholt wind farm"

_Wind Energy Science, 2017_

## Referee Comment (RC1) · Anonymous Referee #1 · 20 Oct 2017

The manuscript presents a comparison between an evaluation of wind turbine SCADA data and mesoscale model simulations for the Anholt wind farm. Assessing wakes in larger wind farms is an important topic that deserves attention. The efficiency of wind farms very much depends on a meaningful consideration of possible wake effects. Although wake properties are very much determined by atmospheric stability, the simulations for this manuscript have been made without taking atmospheric stratification into account.

Unfortunately, I'm inclined to reject the manuscript in its present form. Reasons for this negative decision are:

(1) The Introduction does not present a thorough scientific discussion of the current problems regarding turbine wakes in larger wind farms and does not identify clearly for-

mulated research issues which are to be addressed in this manuscript. The manuscript rather appears to be a collection of isolated evaluations made from the SCADA data, the Jensen park wake model and several mesoscale models (I found "Fuga", a linearized RANS model and WRF mentioned in the text without seeing a clear strategy how and why they have been used).

(2) Page 12, line 2 declares the greatest deficiency of the manuscript: atmospheric stability is not accounted for in the simulations. Why do the authors present such incomplete simulations, although they state in the introduction the importance of atmospheric stability?

(3) The last sentence of the Conclusions gives the final reason why I should not read this paper. Here, the authors clearly state that their results are wind farm specific and SCADA specific and cannot be transferred to other wind farms.

Further issues:

(4) Some references point to grey literature. This is not convenient for the possible reader (e.g., p. 7, line 20).

(5) The denotation of the different wake model simulations is inconsistent. "Park 1" and "Larsen 2" have the same characteristics (as have "Park 2" and "Larsen 1"). This is irritating.

(6) What is meant by a "quadratic sum"? It would be helpful to give a few mathematical formulae in order to avoid unnecessary ambiguity.

(7) The statement in line 20 on p. 2 needs references to the existing literature.

---

## Referee Comment (RC2) · Anonymous Referee #2 · 27 Oct 2017

The paper provides an interesting evaluation of the effect of proximity to the coast on offshore wind farm wake losses which is clearly a relevant and topical area, though there are some points to address: 1) Given that the paper acknowledges that roughness change is the main driver to the change in wind speed offshore, why did the authors not compare the use of WRF with a simple roughness change model to confirm this? 2) It seems strange that a 'full' (non-linearised) RANS model was only used for the southerly flow case. Either such results should be shown for comparison in all cases or not at all. 3) The discrepancy between the RANS model and the results in Fig 7 was put down to a possible prevalence of stable conditions. It was stated that it was not possible to know this, but surely the WRF model results should have given enough information to at least estimate the stability conditions? Although not definitive, this

could lend some weight to this hypothesis. Indeed, in all cases stability is likely to have played a role in wake recovery (and in the coastal transition), though this was not really commented on and would likely have affected the observed wake losses. Also, the RANS model could have been run under stable stratification (perhaps using a couple of z/L scenarios) to test whether a better fit was observed in this case. 4) The results for the capacity factor in 3.2 used the WRF gradient wind speed with wake models. Given that previously, results were presented with both a WRF wind speed gradient and a single representative WRF wind speed, why was this not presented here? 5) The authors suggest that an extension to the work would be to infer the wind speed gradient directly from the SCADA data. It seems odd that this was not already included in this work as it was such an obvious thing to do compared to trying to estimate the effect using a model. I would suggest that it would make this work much stronger if it were included. 6) As suggested in other comments on this paper, the explanation of linear and quadratic wake addition would benefit from some equations and the order of '1' and '2' should be consistent between wake models.

---

## Referee Comment (RC3) · Anonymous Referee #3 · 27 Oct 2017

**Title:** On wake modeling, wind-farm gradients and AEP predictions at the Anholt wind farm

**General comments:**

The authors conducted a large number of simulations using a wide variety of models, and compared simulated values with observations from SCADA. These results can be of use to the scientific community, particularly in regards to the coupling of WRF and wake models, and to the effect of the nearby continent on the wind farm production. However, the abstract, methods, results, and conclusions are not well organized and the reader is left wondering what the real contribution of the work is, and what exactly was done when it comes to specific details of the results and their relevance to the scientific community. The manuscript can be greatly improved by overhauling the organization and text, at which point it can be considered for publication.

**Specific comments:**

Abstract: Very scattered text. Please rewrite. This is very confusing: "*accounting for the horizontal wind-speed gradient gives nearly the same results as averaging all the wake-free wind climates at the turbines' positions or using the wind climate of a position in the middle of the wind farm*". Results of what? AEP? CF? Can you be more direct with the "take home messages" you include in the abstract? This does not belong in the abstract but rather in the discussion section: "*These results are specific for this wind farm, the available dataset, and the derived inflow conditions.*" Can you be quantitative in the abstract, e.g. the model uncertainty is on average x%? What are the relevant results for the greater scientific community? The motivation on page 2, lines 20-26 should be included in a reduced manner in the abstract to give a greater context to why this work is relevant and needed. Below is a rewording that you can use as you rewrite your abstract.

In this work, a wide range of models is used to investigate wake effects at the Anholt offshore wind farm. Undisturbed atmospheric conditions are simulated with WRF for an entire year, and wake effects are simulated with two engineering models (Park and Larsen) and with a linearized Reynolds-Averaged Navier-Stokes solver (Fuga). For the engineering models, linear and quadratic approaches are considered for lateral merging of wake deficits. The effect of the horizontal wind speed gradient over the wind farm on the annual energy production and on the capacity factor is quantified by coupling the WRF and wake models and by comparing the derived predictions to SCADA. Additionally, the ability of the wake models in estimating power losses is evaluated, and the relative uncertainty of each wake model is quantified by bootstrapping the SCADA and to estimate the model-specific error distributions. We find that accounting for the horizontal wind speed gradient is important when estimating the annual energy production but not critical to estimating…? We propose methods for estimating freestream flow conditions based on SCADA, when no measurements are available upstream of the wind farm and quantify their relative performance using the turbines power curve…?

Similarly for the discussion and summary, be more specific with your take home messages. Even after carefully reading the entire manuscript, it is not clear to me by the end what your main results are, and what your contribution is. Results are fragmented and scattered.

"Background" is not a good title for section 2.

Please get rid of "Park1" and "Park2", "Larsen1" and "Larsen2" and choose more descriptive names such as "Park_Linear" and "Park_Quadratic", "Larsen_Linear" and "Larsen_Quadratic".

Remove from all figure captions where you have something like "details in main text".

Be consistent with your verb tenses – either present or past. Example of inconsistency, page 14 line 1: "we use and found".

**Technical corrections:**

| Section | Page/Line | Comment |
|---|---|---|
| Entire manuscript | | Don't hyphenate "wind speed" and "wind direction". You also use hyphens in other various terms that do not call for it, e.g. wind-farm. |

|  |  | Remove the figure references that are left/right and top/bottom and instead use (a), (b), … |
|---|---|---|
| Introduction | 2/22 | "relatively close by" – be quantitative, how many km? |
| Background | 3/9 | Even after being done reading your manuscript, I still don't understand what is the "ensemble" that you are using for your average. Please explain more clearly: is it an ensemble of turbines? Of grid points? Of models? Of runs? |
|  | 3/14-16 | Please give range of wind turbine spacings within the farm, to make it easier for the reader to understand what your model grid spacing means later on. I was left wondering how much spatial interpolation is being done on a 2 km grid, when you place your turbines on the model grid. |
|  | 3/21 | The dataset exclude periods where "any" turbine was parked/idling/etc.? Or only where at least some *n* number of turbines was parked/idling/etc.? |
|  | 22 | The ";" is confusing, please make two sentences there. I don't understand this: "power is 5% above rated power for turbines nr. 1, 36, 65, and 68." |
|  | 23 | How many of these 10-minute time stamps are in 2014, which is the portion you consider in your analysis? |
|  | 26-27 | This is really confusing. Can you have a more lengthy explanation or an equation for u_equivalent? Also, you say how the "inflow reference wind speed" is estimated but what is it defined to be? How about it is defined as … , estimated as … , and used for …? |
|  | 4/Fig. 1 | Can you color the turbines that are used in those groups you define in Tables 1 and 2, to estimate the "inflow wind speed" and direction? Is this what you call the "inflow reference wind speed"? Does "reference" stand for undisturbed, freestream wind speed? |
|  | 4 | Please explain why a group of 4 turbines is used to estimate the wind speed, and only a group of two is used to estimate wind direction? And why are the sectors defined differently? Can you please combine these two tables in one? |
|  | 5/2-12 | How long was the simulation run for? |
|  | 5/10 | Is the model output linearly or logarithmically interpolated to hub height? Please explain. "(the mean wind speed is 9.23 m s–1)" over these sectors or over the entire rose? How does that compare to the "inflow reference wind direction" estimated with your method and your two turbines by region? |
|  | 6/1-6 | Why would you do Park1/Larsen2 for quadratic, and Park2/Larsen1 for linear? Confusing! This entire paragraph is just hard to follow, please rewrite. "We consider three different wake models: the Park wake model with the commonly-used offshore value of k = 0.04; the G. C. Larsen model (Larsen, 2009); and Fuga (Ott et al., 2011). Two methods of laterally merging the wake deficits are considered in the first two models: a linear sum and a quadratic sum." |
|  | 6/7 | What is "a time series basis"? Reword. |
|  | 6/9 | What is a "free" wind speed/direction? Reword. |
|  | 6/11 | "(≈160/340 and 45/235 deg)" confusing – write in words. |
|  | 6/15 | Remove this bit starting with ";for the Anholt…AEP analysis" |
|  | 7/1-12 | These two paragraphs are very confusing. Please rewrite the whole thing, even if you need to be more wordy and/or use equations. |
|  | 7/10 | What is a gradient-based AEP analysis? |
|  | 7/10-12 | I don't understand this last sentence… |
|  | 7/20 | I assume you can reference this pdf in a better way… |
| Results | 7/23 | Why 2014? Why is half of the year in 2013 for which you do have data, ignored here? |
|  | 8/2 | By information you mean the WRF simulated wind direction at hub height? Be specific. |
|  | 8/5-7 | Be quantitative – how small is the effect of the small island relative to the Djursland effect in percentage? |
|  | 8/9 | Everywhere in the manuscript change "all directions" to "omnidirectional" "wind gradient" change to "wind speed gradient" |
|  | 8/10 | How does the magnitude of the WRF gradients compare to those in Paul's RANS work? |
|  | 9/1 | "a effect" change to "an effect" |
|  | 9/1-4 | You need to rewrite this to make it sound a bit more scientific/less speculative. It seems like you are giving a justification for the wind farm wall effect justification for this, but it is poorly worded. Also, this "similar effect" that you are using in your justification is not |

| | | |
|---|---|---|
| | | shown, so maybe say that? |
| | 9/Fig. 5 | Left panel: add small markers to points where each turbine is; Do not connect line as we move from one row to the next (e.g. turbine 30 to 31). Legend… "omnidirectional flow" |
| | 9/7 | "that that" change to "that which"
"assuming a **horizontally** homogeneous" |
| | 9/9 | "highest impact" **of** what **on** what? |
| | 9/6-9 | This sentence is long and confusing. |
| | 9/13 | "larger than 1%" – by how much? |
| | 9/14 | "significant" may be not the best term – is this statistical significant? I'm guessing not. |
| | 9/12-17 | In Section 2 (which may be best called "Methodology") please explain the choices of these turbines #1, #54, #65 in your analysis, as it seems very arbitrary. |
| | 10/1 | Change to "although accounting for the wind farm gradient is important, it does not" |
| | 10/3-4 | This sentence doesn't belong here? |
| | 10/5 | This sentence is too informal, please use scientific writing practices. |
| | 10/8 | By "simulated wind climate" you mean the WRF simulated wind climate? Since you are using so many models, please be very specific when referencing your results. |
| | 10/15-18 | So confusing! Reword. |
| | 10/footnote | I still don't understand what your ensemble is… time series at each wind turbine location? At all the WRF grid points in the innermost domain? |
| | 11/Fig. 6 | Don't use these abbreviations "grad" and "homo" – just spell out the entire term, there is space. What is the SCADA standard "error"? I assume this is the same as "standard deviation" but the term "error" is not usually used in this context, especially when error means something else here (simulations-observations). |
| | 11/1-12 | Why, if the flow is from the west? I don't understand the P3<P31. Is this circling back to your blockage comment earlier on? If so, please remind the reader. |
| | 11/5 | Why this weird number, 168.7? Explain. Be more specific on which information from Table 1 is used, which group? I still don't understand your entire process of estimating these "reference" inflows, when they are used and what for. |
| | 12/3 | Yes you do, you can use WRF output to estimate stability. Please comment on why not do it? |
| | 12/4 | Why is this interesting? Why are the differences so large? |
| | 12/15 | "performing the best" – reword this. |
| | 12/15-17 | Confusing, reword. Why is it not "fair"? Maybe "fair" is not an adequate word here? |
| | 13/3 | Instead of having these numbers in the text can you add them as another column to Table 4, just noting that for PL estimation WRF is not used just the wake models? |
| | 13/7-13 | I'm not sure about this paragraph – it sounds like a justification of your methodology and not really a result. Does it belong elsewhere, maybe Section 2? |
| | 13/17-20 | What does this mean for your analysis? |
| | 14/2-3 | It is counter-intuitive to say that positive values mean under-estimation, so reword this a bit: "where positive $\varepsilon$ values denote a model that overestimates the power (i.e. underestimates the wake loss)" |
| | 14/3 | "mean $<\varepsilon>$ and standard deviation $\sigma\_\varepsilon$ of the distributions" |
| | 14/Table 5 | Get rid of this table and add these numbers to Fig. 9. |
| | 16/1-5 | This paragraph is completely irrelevant. |
| Conclusions | 16/7 | We "confirm" or "reiterate" – you don't really "show" since previous work had already shown this. |
| | | |

---

## Author Comment (AC1) · 7 Dec 2017

Response to the anonymous referee #1:

Thanks a lot for the review. Here our response to the reviewer's comments. The response is given within XXX--- ---XXX

Regards,
The authors

The manuscript presents a comparison between an evaluation of wind turbine SCADA data and mesoscale model simulations for the Anholt wind farm. Assessing wakes in larger wind farms is an important topic that deserves attention. The efficiency of wind farms very much depends on a meaningful consideration of possible wake effects. Although wake properties are very much determined by atmospheric stability, the simulations for this manuscript have been made without taking atmospheric stratification into account. Unfortunately, I'm inclined to reject the manuscript in its present form. Reasons for this negative decision are:
(1) The Introduction does not present a thorough scientific discussion of the current problems regarding turbine wakes in larger wind farms and does not identify clearly formulated research issues which are to be addressed in this manuscript. The manuscript rather appears to be a collection of isolated evaluations made from the SCADA data, the Jensen park wake model and several mesoscale models (I found "Fuga", a linearized RANS model and WRF mentioned in the text without seeing a clear strategy how and why they have been used).

XXX--- We have modified the introduction to highlight the research issues addressed in this manuscript and some of the issues when modeling wakes in large offshore wind farms. As pointed out by the reviewer, we did not explain the motivation of using Fuga and WRF in the introduction of the original submission; we now try to explain why we use them for our analysis (please see the marked-up manuscript which highlights the changes and additions) ---XXX

(2) Page 12, line 2 declares the greatest deficiency of the manuscript: atmospheric stability is not accounted for in the simulations. Why do the authors present such incomplete simulations, although they state in the introduction the importance of atmospheric stability?

XXX--- The westerly and southerly flow cases are presented to show the ability of the wake models to predict the wake loss for particular inflow conditions only. The westerly case being the one where the effect of the land should be the highest and the southerly case being one of those cases with the highest wake losses due to the farm layout. Most of the manuscript is about the wake models being used in a long-time series fashion; it is rather difficult to include atmospheric stability in all the models used here and for the type of use we want them for in this paper, which is the prediction of the AEP. Most important, for the type of the analysis we are focused on (i.e. AEP-like analysis) and given the results we show in terms of AEP/Capacity factor, one can see that inclusion of these effects might not be that important for such analyses. This is because for AEP predictions, the over-and under-estimations we make with these models are generally compensated (unless the long-term atmospheric stability is far from neutral, which is not the case of the North Sea). In the introduction we state that it is important to include them when comparing the wake models for particular flow cases. Also there are no measurements available for stability estimation. Of course WRF provides modelled values of atmospheric stability but using them in a time-series basis is difficult in all wake models and highly uncertain (see Peña and Hahmann, 2012). We now add "We have atmospheric stability measures from the WRF simulations but `instantaneous' WRF stability measures are highly uncertain (Peña and Hahmann, 2012)" ---XXX

(3) The last sentence of the Conclusions gives the final reason why I should not read this paper. Here, the authors clearly state that their results are wind farm specific and SCADA specific and cannot be transferred to other wind farms.

XXX--- We think that it is important to mention that our results are wind-farm specific because they are, as well as most wake evaluations in all literature (if not all). In our particular case, the relative model error is a function of the SCADA and the way SCADA have been treated. This is the reason of our statement. The study shows a way to perform-such an analysis, which can be done in all wind farms but the results are simply only valid for Anholt. It is though probable that the relative differences between models of the relative model error (in Fig. 9 of the original submission) will be similar for different offshore wind farms and configurations, with main differences in the bias from zero relative model error ---XXX

Further issues:
(4) Some references point to grey literature. This is not convenient for the possible reader (e.g., p. 7, line 20).

XXX--- We added this reference following the comments of the associate editor. We now use `proper' referencing in this case ---XXX

(5) The denotation of the different wake model simulations is inconsistent. "Park 1" and "Larsen 2" have the same characteristics (as have "Park 2" and "Larsen 1"). This is irritating.

XXX--- This is now changed as suggested ---XXX

(6) What is meant by a "quadratic sum"? It would be helpful to give a few mathematical formulae in order to avoid unnecessary ambiguity.

XXX--- We now explain with formulae what is meant by linear and quadratic sum as suggested ---XXX

(7) The statement in line 20 on p. 2 needs references to the existing literature.

XXX--- References are added as suggested ---XXX

References:

Peña A. and Hahmann A.N. (2012) Atmospheric stability and turbulence fluxes at Horns Rev – an intercomparison of sonic, bulk and WRF model data. Wind Energ. 15:717—731

---

## Author Comment (AC2) · 7 Dec 2017

Response to the anonymous referee #2:

Thanks a lot for the review. Here our response to the reviewer's comments. The response is given within XXX--- ---XXX

Regards,
The authors

The paper provides an interesting evaluation of the effect of proximity to the coast on offshore wind farm wake losses which is clearly a relevant and topical area, though there are some points to address:

1) Given that the paper acknowledges that roughness change is the main driver to the change in wind speed offshore, why did the authors not compare the use of WRF with a simple roughness change model to confirm this?

XXX--- Although we understand the reviewer's point as it is an interesting comparison, we think that this is out of the scope of the paper and that it will divert the attention of the paper, which is on wake modelling. It will also make the paper much longer and difficult to digest. But for the reviewer's sake we have performed such analysis using the roughness model of WAsP engineering (Astrup et al. 1996) and as observed (qualitatively from the figure below) the model seems to be capable to reproduce the wind speed gradients on the wind farm due to the land nearby. A comparison of WRF and the results of a RANS model are also given in van der Laan et al. (2017a) (this is also now stated when this paper is mentioned in the introduction) ---XXX

[Figure]

Figure. Wind speed simulated at hub height on the Anholt wind farm from a direction of 260 deg using WAsP Engineering.

2) It seems strange that a 'full' (non-linearised) RANS model was only used for the southerly flow case. Either such results should be shown for comparison in all cases or not at all.

XXX--- The RANS simulations are performed for the southerly flow case because, for this particular case, the SCADA does not match well the results of the "simple" wake models as we mention in the original submission (lines 10-12 Page 11). We wanted to investigate if this was because of the simplicity of the wake models. It is however too costly to perform 735 RANS simulations for the westerly flow cases since a single case takes about 3-4 hours using 153 CPUs (8 nodes). We think that is anyway valuable to show that the underestimation of the wake models is not due to the wake models per se but to the way that either data are treated or to conditions that we cannot extract from the SCADA such as atmospheric stability ---XXX

3) The discrepancy between the RANS model and the results in Fig. 7 was put down to a possible prevalence of stable conditions. It was stated that it was not possible to know this, but surely the WRF model results should have given enough information to at least estimate the stability conditions? Although not definitive, this could lend some weight to this hypothesis. Indeed, in all cases stability is likely to have played a role in wake recovery (and in the coastal transition), though this was not really commented on and would likely have affected the observed wake losses. Also, the RANS model could have been run under stable stratification (perhaps using a couple of z/L scenarios) to test whether a better fit was observed in this case.

XXX--- For the southerly flow case, the WRF simulations show a wide range of atmospheric stability conditions and in `average' the atmosphere is actually close to neutral following the derived stability estimates from WRF. As we now mention in the revised paper, the instantaneous estimations of stability from WRF are rather uncertain (Pena and Hahmann, 2012). We acknowledge that it would be great to include RANS results with stability. However, our RANS model has only been validated with measurements and LES for neutral conditions (van der Laan et al., 2015a; van der Laan et al., 2015b). We have two published (Koblitz et al., 2015; van der Laan et al., 2017b) and one unpublished methods to account for atmospheric stability in RANS, which are all not yet validated to be used for wakes. We are currently conducting this research in a dedicated paper ---XXX

4) The results for the capacity factor in 3.2 used the WRF gradient wind speed with wake models. Given that previously, results were presented with both a WRF wind speed gradient and a single representative WRF wind speed, why was this not presented here?

XXX--- Given that the capacity factor is directly related to the AEP, one can estimate the difference in capacity factors when using different types of wind information by using the differences in AEP shown in table 3 of the original submission ---XXX

5) The authors suggest that an extension to the work would be to infer the wind speed gradient directly from the SCADA data. It seems odd that this was not already included in this work as it was such an obvious thing to do compared to trying to estimate the effect using a model. I would suggest that it would make this work much stronger if it were included.

XXX--- We guess the reviewer refers here to the last two sentences of the first paragraph of the original Discussion. In fact, this is more complicated than it sounds because we need to find all instances where all undisturbed turbines are concurrently operating showing quality-checked and calibrated yaw positions, power and pitch values. Also, and probably more difficult is it to define exactly what is a turbine under undisturbed inflow conditions when you have directions of all 111 individual turbines. This is partly the reason of the use of WRF since we can extract all 111 wake-free wind climates. The results in Fig. 5-right show that for most westerly directions, where the horizontal wind-speed gradient is the highest, WRF seems to do a fairly good job compared to individually-derived wind speeds for the most westerly row. We

have reformulated the sentences to clarify the aspects of such analysis and added a footnote to clarify that the wind-speed gradient cannot be infer from wake-affected turbines ---XXX

6) As suggested in other comments on this paper, the explanation of linear and quadratic wake addition would benefit from some equations and the order of '1' and '2' should be consistent between wake models.

XXX--- We have done this as suggested ---XXX

References

Astrup P., Jensen N.O. and Mikkelsen T. (1996) Surface roughness model for LINCOM. Risø-R-900(EN). Risø National Laboratory, Roskilde, Denmark

van der Laan M.P., Peña A., Volker P., Hansen K.S., Sørensen N.N., Ott S. and Hasager C.B. (2017a) Challenges in simulating coastal effects on an offshore wind farm. J. Phys.: Conf. Ser., 854, 012046

Peña A. and Hahmann A.N. (2012) Atmospheric stability and turbulence fluxes at Horns Rev – an intercomparison of sonic, bulk and WRF model data. Wind Energ. 15:717—731

van der Laan, M. P., Sørensen, N. N., Réthoré, P.-E., Mann, J., Kelly, M. C., Troldborg, N., Schepers, J. G., and Machefaux, E. (2015) An improved k-ε model applied to a wind turbine wake in atmospheric turbulence. *Wind Energy, 18*, 889-489

van der Laan, M. P., Sørensen, N. N., Réthoré, P.-E., Mann, J., Kelly, M. C., Troldborg, N., Hansen, K. S. and Murcia, J. P. (2015) The k-ε-fP model applied to wind farms. *Wind Energy, 18*, 2065-2084

Koblitz, T., Bechmann, A., Sogachev, A., Sørensen, N. and Réthoré, P.-E. (2015) Computational Fluid Dynamics model of stratified atmospheric boundary-layer flow. *Wind Energy, 18*, 75-89

van der Laan, M. P.; Sørensen, N. N. & Kelly, M. C. (2017b) A new k-epsilon model consistent with Monin-Obukhov similarity theory. *Wind Energy, 20*, 479-489

---

## Author Comment (AC3) · 7 Dec 2017

Response to the anonymous referee #3:

Thanks a lot for the review. Here our response to the reviewer's comments. The response is given within XXX--- ---XXX

Regards,
The authors

**General comments:**
The authors conducted a large number of simulations using a wide variety of models, and compared simulated values with observations from SCADA. These results can be of use to the scientific community, particularly in regards to the coupling of WRF and wake models, and to the effect of the nearby continent on the wind farm production. However, the abstract, methods, results, and conclusions are not well organized and the reader is left wondering what the real contribution of the work is, and what exactly was done when it comes to specific details of the results and their relevance to the scientific community. The manuscript can be greatly improved by overhauling the organization and text, at which point it can be considered for publication.

**Specific comments:**
Abstract: Very scattered text. Please rewrite. This is very confusing: "*accounting for the horizontal wind-speed gradient gives nearly the same results as averaging all the wake-free wind climates at the turbines' positions or using the wind climate of a position in the middle of the wind farm*". Results of what? AEP? CF? Can you be more direct with the "take home messages" you include in the abstract?

XXX--- The abstract has largely been changed taking the suggestions of the reviewer. The changes and additions can be clearly seen in the marked-up version of the manuscript ---XXX

This does not belong in the abstract but rather in the discussion section: "*These results are specific for this wind farm, the available dataset, and the derived inflow conditions.*"

XXX--- Given that we provide quantitative results, we think that is very important to say that the numbers are specific for this wind farm, these inflow conditions and this dataset---XXX

Can you be quantitative in the abstract, e.g. the model uncertainty is on average x%? What are the relevant results for the greater scientific community?

XXX--- See our previous two responses ---XXX

The motivation on page 2, lines 20-26 should be included in a reduced manner in the abstract to give a greater context to why this work is relevant and needed. Below is a rewording that you can use as you rewrite your abstract.

In this work, a wide range of models is used to investigate wake effects at the Anholt offshore wind farm. Undisturbed atmospheric conditions are simulated with WRF for an entire year, and wake effects are simulated with two engineering models (Park and Larsen) and with a linearized Reynolds-Averaged Navier-Stokes solver (Fuga). For the engineering models, linear and quadratic approaches are considered for lateral merging of wake deficits. The effect of the horizontal wind speed gradient over the wind farm on the annual energy production and on the capacity factor is quantified by coupling the WRF and wake models and by comparing the derived predictions to SCADA. Additionally, the ability of the wake models in estimating power losses is evaluated, and the relative uncertainty of each wake model is quantified by bootstrapping the SCADA and to estimate the model-specific error distributions. We find that accounting for the horizontal wind speed gradient is important when estimating the annual energy production but not critical to estimating…? We propose methods for estimating freestream flow conditions based on SCADA, when no measurements are available upstream of the wind farm and quantify their relative performance using the turbines power curve…?

XXX--- We appreciate the suggestion of abstract by the reviewer. We now use some of the suggestion to write a revised abstract with what we consider has a better flow. We also add some of the text regarding motivation as suggested ---XXX

Similarly for the discussion and summary, be more specific with your take home messages. Even after carefully reading the entire manuscript, it is not clear to me by the end what your main results are, and what your contribution is. Results are fragmented and scattered.

XXX--- We think this was partly because of the way the abstract and introduction were written and also because of the rather `disruptive' last paragraph in the original discussion. We have removed this last paragraph. We think the abstract and conclusions provide with the important take home messages; a sentence has been added to the second paragraph of the conclusions to link the results for the individual flow cases with the overall power loss. ---XXX

"Background" is not a good title for section 2.

XXX--- We change it for "Methods" ---XXX

Please get rid of "Park1" and "Park2", "Larsen1" and "Larsen2" and choose more descriptive names such as "Park_Linear" and "Park_Quadratic", "Larsen_Linear" and "Larsen_Quadratic".

XXX--- This is now changed as suggested by the reviewers ---XXX

Remove from all figure captions where you have something like "details in main text".

XXX--- Removed as suggested by the reviewer ---XXX

Be consistent with your verb tenses – either present or past. Example of inconsistency, page 14 line 1: "we use and found"

XXX--- We have gone through the paper to find such inconsistencies ---XXX

**Technical corrections:**

| Section | Page/Line | Comment |
|---|---|---|
| Entire manuscript | | Don't hyphenate "wind speed" and "wind direction". You also use hyphens in other various terms that do not call for it, e.g. wind-farm. |
| | | XXX—hyphenation is a matter of style and we think that it is the editor who decides whether this is appropriate. You will not find (if you do it is a typo) two isolated words hyphenated, e.g. wind-farm but wind-farm gradients ---XXX |
| | | Remove the figure references that are left/right and top/bottom and instead use (a), (b), … |
| | | XXX—This is the style we use and we have used it previously in other publications in the same journal ---XXX |
| Introduction | 2/22 | "relatively close by" – be quantitative, how many km? |
| | | XXX—We add the number as recommended ---XXX |
| Background | 3/9 | Even after being done reading your manuscript, I still don't understand what is the "ensemble" that you are using for your average. Please explain more clearly: is it an ensemble of turbines? Of grid points? Of models? Of runs? |
| | | XXX--- Since it is not necessary that the values that we average are equally separated in a time-series form, we clarify that these averages are ensemble averages. In the particular case of Eqn. (2) of the original submission it is an average of power values ---XXX |
| | 3/14-16 | Please give range of wind turbine spacings within the farm, to make it easier for the reader to understand what your model grid spacing means later on. I was left wondering how much spatial interpolation is being done on a 2 km grid, when you place your turbines on the model grid. |
| | | XXX--- We add "The smallest distance between the turbines is 4.9 rotor diameters"---XXX |

| 3/21 | The dataset exclude periods where "any" turbine was parked/idling/etc.? Or only where at least some *n* number of turbines was parked/idling/etc.? |
|---|---|
| | XXX--- "any" is added as suggested ---XXX |
| 22 | The ";" is confusing, please make two sentences there. I don't understand this: "power is 5% above rated power for turbines nr. 1, 36, 65, and 68." |
| | XXX--- We now split the sentence into two and reformulated the last part to avoid confussion ---XXX |
| 23 | How many of these 10-minute time stamps are in 2014, which is the portion you consider in your analysis? |
| | XXX--- If the type of analysis is performed with the filtered SCADA time series, then all the time series is considered (not only 2014) except for the results regarding the capacity factor, in which we use all non-filtered SCADA for 2014 as stated in the section "Capacity factor" ---XXX |
| 26-27 | This is really confusing. Can you have a more lengthy explanation or an equation for u_equivalent? Also, you say how the "inflow reference wind speed" is estimated but what is it defined to be? How about it is defined as … , estimated as … , and used for …? |
| | XXX--- We have reformulated these sentences and provided an extended explanation of the equivalent wind speed ---XXX |
| 4/Fig. 1 | Can you color the turbines that are used in those groups you define in Tables 1 and 2, to estimate the "inflow wind speed" and direction? Is this what you call the "inflow reference wind speed"? Does "reference" stand for undisturbed, freestream wind speed? |
| | XXX--- Colors are added as suggested. As it is stated in line 27/p 3 of the original submission, the inflow reference speed is estimated from wake-free groups of turbines, so yes, it is an undisturbed freestream speed ---XXX |
| 4 | Please explain why a group of 4 turbines is used to estimate the wind speed, and only a group of two is used to estimate wind direction? And why are the sectors defined differently? Can you please combine these two tables in one? |
| | XXX--- The two tables are now combined as suggested. We have extended the explanation of the computation of the inflow wind direction as suggested ---XXX |
| 5/2-12 | How long was the simulation run for? |
| | XXX--- The simulations were originally performed for another project and are described in detail in the reference we provide in the text. For the reviewer's knowledge, the model was run during nearly 4 months and is a 30-year mesoscale model simulation ---XXX |
| 5/10 | Is the model output linearly or logarithmically interpolated to hub height? Please explain. "(the mean wind speed is 9.23 m s−1)" over these sectors or over the entire rose? How does that compare to the "inflow reference wind direction" estimated with your method and your two turbines by region? |
| | XXX--- We add "The model output is logarithmically interpolated to hub height" as suggested. The mean wind speed is an all-sector mean wind speed so this information is now added. It is not important how well the simulated mean wind speed compares to that estimated by us from the SCADA since the latter is less than an ideal time series due to the filtering we apply (described in the SCADA section) ---XXX |
| 6/1-6 | Why would you do Park1/Larsen2 for quadratic, and Park2/Larsen1 for linear? Confusing! This entire paragraph is just hard to follow, please rewrite. "We consider three different wake models: the Park wake model with the commonly-used offshore value of k = 0.04; the G. C. Larsen model (Larsen, 2009); and Fuga (Ott et al., 2011). Two methods of laterally merging the wake deficits are considered in the first two models: a linear sum and a quadratic sum." |
| | XXX--- We have changed the names of Park 1/2 and Larsen 1/2 to linear and quadratic to avoid confusion as suggested and we also take the suggestion of the reviewer regarding the paragraph ---XXX |
| 6/7 | What is "a time series basis"? Reword. |

| | | XXX--- This has been reworded as suggested ---XXX |
|---|---|---|
| | 6/9 | What is a "free" wind speed/direction? Reword. |
| | | XXX--- Changed to "undisturbed" ---XXX |
| | 6/11 | "(≈160/340 and 45/235 deg)" confusing – write in words. |
| | | XXX--- We do not really understand why is this confusing but we now use more words anyway ---XXX |
| | 6/15 | Remove this bit starting with ";for the Anholt…AEP analysis" |
| | | XXX--- Removed as suggested ---XXX |
| | 7/1-12 | These two paragraphs are very confusing. Please rewrite the whole thing, even if you need to be more wordy and/or use equations. |
| | | XXX--- We have rewritten both paragraphs, in particular the first one, which is the one providing the details of how we account for the horizontal wind gradient. Here it is also now defined what a gradient-based analysis is ---XXX |
| | 7/10 | What is a gradient-based AEP analysis? |
| | | XXX--- See our previous response ---XXX |
| | 7/10-12 | I don't understand this last sentence… |
| | | XXX--- He have also rewritten this sentence so that it is clear what we mean with calculations using pre-computed LUTs ---XXX |
| | 7/20 | I assume you can reference this pdf in a better way… |
| | | XXX--- Corrected as suggested ---XXX |
| Results | 7/23 | Why 2014? Why is half of the year in 2013 for which you do have data, ignored here? |
| | | XXX--- It is simply to have a complete year and not bias the AEP estimation ---XXX |
| | 8/2 | By information you mean the WRF simulated wind direction at hub height? Be specific. |
| | | XXX--- We replace "information" by "simulated wind direction at hub height" as suggested ---XXX |
| | 8/5-7 | Be quantitative – how small is the effect of the small island relative to the Djursland effect in percentage? |
| | | XXX--- We have added a sentence with numbers regarding the differences between the influence of both land bodies on the farm ---XXX |
| | 8/9 | Everywhere in the manuscript change "all directions" to "omnidirectional"
"wind gradient" change to "wind speed gradient" |
| | | XXX--- Corrected as suggested ---XXX |
| | 8/10 | How does the magnitude of the WRF gradients compare to those in Paul's RANS work? |
| | | XXX--- For the reviewer's knowledge: WRF and RANS predict comparable trends of the velocity gradient with respect to wind direction. However, the gradient calculated by WRF is more wide spread with respect to the RANS results (see van der Laan et al., 2017) ---XXX |
| | 9/1 | "a effect" change to "an effect" |
| | | XXX--- Corrected as suggested ---XXX |
| | 9/1-4 | You need to rewrite this to make it sound a bit more scientific/less speculative. It seems like you are giving a justification for the wind farm wall effect justification for this, but it is poorly worded. Also, this "similar effect" that you are using in your justification is not shown, so maybe say that? |
| | | XXX--- We add "(not shown)" as suggested and use some rewording to sound less speculative as recommended ---XXX |
| | 9/Fig. 5 | Left panel: add small markers to points where each turbine is; Do not connect line as we move from one row to the next (e.g. turbine 30 to 31). Legend… "omnidirectional flow" |
| | | XXX--- Changed as suggested ---XXX |
| | 9/7 | "that that" change to "that which"
"assuming a **horizontally** homogeneous" |
| | | XXX--- Changed as suggested ---XXX |
| | 9/9 | "highest impact" **of** what **on** what? |
| | | XXX--- Changed to "difference" and so it is self-explanatory ---XXX |
| | 9/6-9 | This sentence is long and confusing. |
| | | XXX--- We slightly reword and shorten the sentence as suggested ---XXX |

| | |
|---|---|
| 9/13 | "larger than 1%" – by how much? |
| | XXX--- We provide later (line 14 page 9 of the original submission) the AEP reference value |
| 9/14 | "significant" may be not the best term – is this statistical significant? I'm guessing not. |
| | XXX--- Changed to "large" in two instances as suggested---XXX |
| 9/12-17 | In Section 2 (which may be best called "Methodology") please explain the choices of these turbines #1, #54, #65 in your analysis, as it seems very arbitrary. |
| | XXX--- We do not think that it seems arbitrary. As the original submission states in lines 13, 15 and 17 page 9, these turbines are chosen either because of their strategic location or because in case of 1 the wind speed is the lowest observed---XXX |
| 10/1 | Change to "although accounting for the wind farm gradient is important, it does not" |
| | XXX--- Changed as suggested ---XXX |
| 10/3-4 | This sentence doesn't belong here? |
| | XXX--- This value is here provided so that the reader can see how different the wind of turbines nr. 1, 54 and 65 is compared to the average homogenous wind  ---XXX |
| 10/5 | This sentence is too informal, please use scientific writing practices. |
| | XXX--- We have rewritten the sentence as suggested ---XXX |
| 10/8 | By "simulated wind climate" you mean the WRF simulated wind climate? Since you are using so many models, please be very specific when referencing your results. |
| | XXX--- We have added "WRF" as suggested ---XXX |
| 10/15-18 | So confusing! Reword. |
| | XXX--- The sentence has been split into two and reworded ---XXX |
| 10/footnote | I still don't understand what your ensemble is… time series at each wind turbine location? At all the WRF grid points in the innermost domain? |
| | XXX--- The footnote has been removed---XXX |
| 11/Fig. 6 | Don't use these abbreviations "grad" and "homo" – just spell out the entire term, there is space. What is the SCADA standard "error"? I assume this is the same  as  "standard deviation" but the term "error" is not usually used in this context, especially when error means something else here (simulations-observations). |
| | XXX--- We changed to "standard error of the mean" which is equal to sigma/\sqrt{n}, with n being the number of samples. We also avoid the abbreviations as suggested ---XXX |
| 11/1-2 | Why, if the flow is from the west? I don't understand the P3<P31. Is this circling back to your blockage comment earlier on? If so, please remind the reader. |
| | XXX--- There are couple of possible reasons: first it is a large wind sector, second distances are large between these two archs and so wakes are small , third the wake meanders, and fourth the inflow is not uniform ---XXX |
| 11/5 | Why this weird number, 168.7? Explain. Be  more  specific on  which  information from Table 1 is used, which group? I still don't understand your entire process of estimating these "reference" inflows, when they are used and what for. |
| | XXX--- We now add ", which is the direction where turbines nr. 45 and 46 are aligned". We also rephrased the text so that it reads "that are derived from the SCADA of turbines nr. 45 and 67—68 (Table 1)" to be more explicit---XXX |
| 12/3 | Yes you do, you can use WRF output to estimate stability. Please comment on why not do it? |
| | XXX--- We do not have observations of atmospheric stability. We add "We have atmospheric stability measures from the WRF simulations but `instantaneous' WRF stability measures are highly uncertain (Peña and Hahmann, 2012)" ---XXX |
| 12/4 | Why is this interesting? Why are the differences so large? |
| | XXX--- We delete "interesting" from the sentence. As we mention, the period is different from that used by Nygaard (2014) ---XXX |
| 12/15 | "performing the best" – reword this. |
| | XXX--- See next response ---XXX |

| | 12/15-17 | Confusing, reword. Why is it not "fair"? Maybe "fair" is not an adequate word here? |
|---|---|---|
| | | XXX--- We reword the sentences as suggested: "However, it is important to note that wind turbines are not always working and underperform when compared to the manufacturer's power curve. The predicted AEP/capacity factor of a combined mesoscale-wake model is typically lower than the observed value; however, we want to know the capacity factor of a wind farm regardless of the operating conditions."---XXX |
| | 13/3 | Instead of having these numbers in the text can you add them as another column to Table 4, just noting that for PL estimation WRF is not used just the wake models? |
| | | XXX--- We think that this will be confusing as WRF is not used for the PL estimations and because these are two different datasets --- XXX |
| | 13/7-13 | I'm not sure about this paragraph – it sounds like a justification of your methodology and not really a result. Does it belong elsewhere, maybe Section 2? |
| | | XXX--- In section 2 we do not show any results that involve the analysis of SCADA so we choose this place as the power loss is directly dependent on the derived undisturbed inflow conditions ---XXX |
| | 13/17-20 | What does this mean for your analysis? |
| | | XXX--- We respond to this question in the first paragraph of the original discussion ---XXX |
| | 14/2-3 | It is counter-intuitive to say that positive values mean under-estimation, so reword this a bit: "where positive ε values denote a model that overestimates the power (i.e. underestimates the wake loss)" |
| | | XXX--- Changed as suggested ---XXX |
| | 14/3 | "mean $<\varepsilon>$ and standard deviation $\sigma_{\varepsilon}$ of the distributions" |
| | | XXX--- Changed to "The mean and standard deviation of the distributions of $\epsilon$, $<epsilon>$ and $\sigma_{\epsilon}$" ---XXX |
| | 14/Table 5 | Get rid of this table and add these numbers to Fig. 9. |
| | | XXX--- Corrected as suggested ---XXX |
| | 16/1-5 | This paragraph is completely irrelevant. |
| | | XXX--- Removed as suggested ---XXX |
| Conclusions | 16/7 | We "confirm" or "reiterate" – you don't really "show" since previous work had already shown this. |
| | | XXX--- Changed to "confirm" as suggested ---XXX |

References:

Peña A. and Hahmann A.N. (2012) Atmospheric stability and turbulence fluxes at Horns Rev – an intercomparison of sonic, bulk and WRF model data. Wind Energ. 15:717—73

van der Laan M.P., Peña A, Volker P., Hansen K.S., Sørensen N.N., Ott S., and Hasager C.B. (2017) Challenges in simulating coastal effects on an offshore wind farm. J. Phys.: Conf. Series 854, 012046

---

## Referee Report (RR1)

[referee-annotated manuscript omitted]

---

## Author Response (AR2)

Response to the anonymous referee #1:

Thanks a lot for the review. Here our response to the reviewer's comments. The response is given within XXX--- ---XXX

Regards,
The authors

Dear authors, thank you for revising your manuscript. It has improved considerably.  I recommend to accept the manuscript subject to the insertion of two further references:

Page 2, lines 31 - 35: please have a look at: "Dörenkämper, M., M. Optis, A. Monahan, G. Steinfeld: On the Offshore Advection of Boundary-Layer Structures and the Influence on Offshore Wind Conditions. Bound.-Lay. Meteorol., 155, 459–482 (2015)" and make proper reference to this paper. It describes the influence of onshore boundary-layer features on offshore wind parks.

XXX--- We were not aware of this very interesting work, which is now cited in the introduction---XXX

Page 3, lines 1 - 8: please have a look at: "Andreas Platis, Simon. K. Siedersleben, Jens Bange, Astrid Lampert, Konrad Bärfuss, Rudolf Hankers, Beatriz Cañadillas, Richard Foreman, Johannes Schulz-Stellenfleth, Bughsin Djath, Thomas Neumann, Stefan Emeis, First in situ evidence of wakes in the far field behind offshore wind farms, Scientific Reports, 2018, doi 10.1038/s41598-018-20389-y" and make proper reference to this paper. This paper describes a successful effort to model an offshore wind park with WRF. The model results are confirmed by in situ aircraft measurements for the first time.

XXX--- The reference is now included as suggested ---XXX

Response to the anonymous referee #3:

Thanks a lot for the review. Here our response to the reviewer's comments. The response is given within XXX--- ---XXX

Regards,
The authors

This is a much improved version. The work performed and results (and their relevance) are much more clearly presented. Please address the technical corrections marked up in the PDF (to modify text, a table and a figure) and the few comments about the actual content (minor comments, which can be easily addressed and make your results more valuable).

XXX---- We have taken into account most of the comments addressed by the reviewer in the pdf. Some we do not think will improve the paper, specifically:
- "Reynolds-average" should have "average" in lower case although the acronym is RANS (the reviewer wants to have capital letter)
- Allowance of footnotes should be a concern of the manuscript technical editors
- In Fig. 4 we need to have different colorbars because when the same is used the clear horizontal gradient disappears either on the left or on the right plot
- The reviewer wants to know why the Park-quadratic and Fuga relative errors are so similar. We do not know and this might not happen for other wind farm. It is clear that if we use another wake decay coefficient the results of the two models will differ more so it is just coincidental. Park-linear is also very different from those two results.
- The reviewer suggests adding "as already shown in Fig. 7" when we discuss what will be the result of extrapolating the relative model errors to an AEP analysis. Figure 7 is only a flow case and does not necessarily show the general trend of the models when performing AEP analysis.
---XXX

The only remaining major point is to add more of a discussion to the Conclusions section, which is now a summary of what was presented in the results. Namely, why do these models perform so differently? Was this what you expected? If not, what did you expect? Why doesn't the RANS model perform better since it has more physics in it? Can you postulate some reasons for it? What should the research community do if they were to pick up your work and give continuity to it? Some main future directions (besides applying your methods to other sites) and questions still left unanswered would be very valuable.

XXX--- We like to have the conclusion section as it is, i.e., with only concluding remarks about the results we show. We anyway added a last paragraph to the discussion section where we address the questions/discussions raised by the reviewer. ---XXX

After these minor topics are addressed and the Conclusions section improved, I support publication of the manuscript which presents novel work and valuable results.

[revised manuscript text omitted]